# Effects of a Mentor Program for Coaches on the Coach-Athlete Relationship

**DOI:** 10.3390/sports9080116

**Published:** 2021-08-23

**Authors:** Jan Arvid Haugan, Frode Moen, Maja Olsen Østerås, Frode Stenseng

**Affiliations:** 1Department of Education and Lifelong Learning, Norwegian University of Science and Technology, 7491 Trondheim, Norway; frode.moen@ntnu.no (F.M.); frode.stenseng@ntnu.no (F.S.); 2Centre for Elite Sports Research, Department of Neuromedicine and Movement Science, Norwegian University of Science and Technology, 7491 Trondheim, Norway; Maja.OlsenOsteras@olympiatoppen.no

**Keywords:** coach education, control-experiment study, communication skills

## Abstract

The present study was designed to explore the effects of a one-year coach education program on coaches’ perceptions of their communication skills and co-orientation of their coach-athlete relationships. The study was designed with an experimental group and a control group. The experiment group consisted of 66 coaches (and 295 athletes) who received formal mentoring and the control group consisted of 41 coaches (and 148 athletes) who did not receive any mentoring. Data were analysed using structural equation modelling with autoregressive cross-lagged analysis. Results from the self-reported questionnaire at pre-test and post-test showed that the reciprocity of the coach-athlete relationships was not statistically significant. However, coaches’ experience of change in attention skills from the pre-test to the post-test positively predicted changes in their own perception of the coach-athlete relationship, whereas this association was not significant in the athletes’ perceptions. Moreover, the coach education programme increased coaches’ perception of their relational bonds with their athletes, but this increase did not correspond with an increase in athletes’ perception of the relational bonds with their coach. Practical implications and suggestions for further research are discussed in light of these findings.

## 1. Introduction

The overall purpose of sports coaching is to facilitate athletes’ performance development and successful outcomes seem to be considerably dependent on the knowledge, skills and competence of the respective coach in specific domains [1,2]. In their in-depth analysis of what constitutes coaching effectiveness, Côté and Gilbert [3] pointed out three knowledge domains that constitute coaching expertise: (1) *Professional* (content knowledge of sport science and how he/she teaches sports skills), (2) *Intrapersonal* (knowledge of how a coach becomes aware and reflective) and (3) *Interpersonal* (knowledge of how to connect to others such as players, the media and other coaches). It is the last domain, interpersonal knowledge, and how a one-year coach development program impacts the coach-athlete relationship that is of special interest in this article.

Research findings indicate that coaches’ learning and development can be facilitated through both *formal* sources (institutional education programs with academic quality criteria) and *non-formal* sources (outside formal academic systems and independent of bounded curriculum and formulated quality criteria) [4,5]. Other research suggests that it is through non-formal sources such as observations of and dialogue with other peers, and reflections based on personal practical experiences, that the interpersonal knowledge among coaches primarily grows and develops [6,7]. Hence, in addition to formal learning sources, a coach education program has to include non-formal learning activities [8,9].

Mentoring from other experienced coaches is a source that can be both formal and non-formal [10,11], it bridges the gap between theory and practice by the potential learning opportunities that arise from professional experience [12,13]. Based on this, mentoring has been advocated as a method for facilitating coach learning through guidance, observation and reflective practice that enables coaches to acquire and deal with the complexity of sports coaching [14,15]. However, a systematic review that aimed to highlight gaps in the knowledge and research agenda by Leeder and Sawiuk [16] to update the work of Jones et al. [12], found little evidence of associations between mentoring and changes in coaching practice. Their review of contemporary trends within the sports coach mentoring literature concludes that “the impact and evaluation of formal mentoring programmes are worthy of increased consideration” [16]. More specifically, and in line with Bloom [17], they argue there is a lack of evaluation-based research with indicators of effectiveness. This includes research on the effects on the coach-athlete relationship.

### 1.1. The Coach-Athlete Relationship

The coach-athlete relationship lies at the heart of effective coaching [18,19]. Jowett [20] claims that this dyad is “… the medium that motivates, assures, satisfies, comforts, and supports coaches and athletes to enhance their sport experience, performance, and well-being” (p. 154), and that “… the quality of the relationship can function as a barometer of coaching effectiveness” (p. 156). In Jowett’s [14,21] 3 + 1 Cs model of the coach-athlete relationship, *closeness* (i.e., an emotional connection reflected in trust and respect), *commitment* (i.e., motivation to maintain a close relationship over time) and *complementarity* (i.e., behaviour reflected in interactions that are responsive, relaxed and friendly) constitute the first three Cs. The last C, *co-orientation*, refers to the degree of interconnection between the coach’s and athlete’s perception of the relational quality. From this perspective, the quality of the coach-athlete relationship is dependent on the *reciprocity* of trust, motivation and behaviour between the coach and the athlete [9,22]. 

Research on the coach-athlete relationship has traditionally focused solely on the athletes’ perceptions of relational qualities [23,24], or solely on the coaches’ perceptions [25,26]. However, recent relationship research has underlined the need for a dyadic, rather than individual, level of analysis. This has led to studies that have illuminated the reciprocity of the coach-athlete relationship [20,27]. Here, the central notion is that a high-quality relationship has to be mutual and aligned from both the coaches’ and athletes’ perspectives with Jowett’s 3 + 1 Cs model. Studies have found that reciprocal perceptions of a strong relational bond between coach and athlete lead to more satisfaction with performance [28,29], higher levels of motivation [30,31], and physical self-concept [32]. In addition, there are higher levels of collective efficacy [33,34]. The negative effects of low-quality coach-athlete relationships include interpersonal conflict [35,36] and athlete burnout [37,38].

Among the pioneering work on the reciprocity of coach-athlete dyads was a study by Stebbings, Taylor and Spray [39] that explored eighty-two coaches’ interpersonal behaviour as a mechanism for well-being and ill-being contagion from coach to athlete and vice versa. They found that the relationships were mediated by athletes; perceptions of their coaches; interpersonal style and that the two perceptions of relational quality were not reciprocal. Other research presents mixed findings. Some studies found that the reciprocity in perceptions of the coach-athlete relationship predicted need satisfaction for both coaches and athletes [27,40], while others found that there are significant differences between coaches’ and athletes’ perceptions of the motivational climate, for instance [41,42].

In sports coaching, especially when working with young athletes, it is the coach who has the main responsibility for the establishment, maintenance, development and repair of the coach-athlete relationship [3,43]. Among the mechanisms that affect and are affected by this dyad, interpersonal communication skills seem to be a key factor [21,40].

### 1.2. Interpersonal Communication Skills in Sports Coaching

In the context of sports coaching, Gilbert [1] claims that “the most effective coaching strategy for building and sustaining a quality coach-athlete relationship is communication” (p. 78). This is supported by research in other contexts related to the facilitation of growth and learning such as education [44], leadership [45] and counselling [46]. Other studies indicate that coaching efficiency and relational qualities are dependent on the coach’s attention skills and influencing communication skills [47,48].

Attention skills encompass both respectful, responsive behaviour and the ability to listen with full attention in a manner that gives the athlete the impression of being seen, heard and understood [49,50]. These skills seem crucial as they build trust and feelings of safety in the relationship. They also enable the athlete to be involved in dialogue and scrutinize their personal development and growth [46,51]. Influencing skills are increasingly important as the coaching process progresses after the coach-athlete relationship is established and developed through the coach’s use of attention skills [2,52]. These skills encompass the use of questions that influence the athletes’ motivation, behaviour and awareness which are necessary to discover new perspectives on his or her performance development.

In sum, the coaches’ attention skills and influencing skills seem decisive for building high-quality coach-athlete relationships in sports [48,53]. However, research indicates that coaches over- and under-report their own interpersonal communication skills [27,41], and other studies point out the need for coaches to have perceptual distance and feedback concerning their own behaviour [42,54]. These findings underline the need for interpersonal coach education interventions. Despite this, research reveals that formal academic education programs often lack interpersonal coach education interventions [5,55].

### 1.3. The Present Study

The present study was designed to explore the effects on the coaches perceptions of their communication skills and the co-orientation of the interpersonal relationship between the coaches and their athletes during a one-year coach education program. Based on our findings from the literature review, three hypotheses were formulated:

**Hypothesis** **1.** 
*A coach education program based on formal mentoring affects both coaches’ and athletes’ perceptions of their relational bonds at the post-test (T2) [4,5].*


**Hypothesis** **2.** 
*Coaches’ and athletes’ agree on their perceptions of their relational bonds at both the pre-test (T1) and the post-test (T2) [27,38].*


**Hypothesis** **3.** 
*The coaches’ perceptions of changes in their attention skills and influencing skills from the pre-test to the post-test (T2–T1) affect both coaches’ and athletes’ perceptions of their relational bonds at the post-test (T2) [47,48].*


## 2. Method

### 2.1. Procedure and Participants

This study is based on a two-year coaching education program initiated by the Norwegian Olympic Sports Centre (NOSC). This national organization is part of the main body responsible for the political government of all sports in Norway: the Norwegian Olympic and Paralympic Committee and Confederation of Sports. NOSC has the authority and responsibility for performance development among Norwegian elite sports, including the training and management of elite coaches and athletes. The participants in this study were recruited from the two-year coaching education program initiated and arranged by the NOSC for promising coaches and their ambitious athletes.

### 2.2. The Coach Education Program—Formal Mentoring of Coaches

The two-year coach education program was organized as individual mentoring of coaches led by twenty-six people with elite sports expertise and mentoring education. The purpose of the coach education program is to nurture and develop coaching skills and knowledge through guidance and facilitation of reflection based on practical experience during the coaching of athletes.

In parallel with the first year of the coach education program, all mentors had to complete a mentoring education program arranged by the Department of Education and Life-Long Learning at the Norwegian University of Science and Technology. This mentor education program consisted of four gatherings (each lasting for two days) and individual lessons that the mentors had to complete between gatherings. The education program was completed with a final written exam and offered 7.5 university credits. The mentor education program was designed to give the mentors, theoretical, practical and research-based knowledge about mentoring based on a coach-centred approach [56]. This implies an emphasis on the improvement of the mentors’ relationship with their mentored coaches through improved attention skills (e.g., using open-ended questions, stimulating reflections based on the coaches’ own experience and listening skills) and influencing skills (e.g., questions used to influence motivation, behaviour and awareness). The primary outcome of the mentor program was to facilitate developmental processes for the mentored coaches in the coach education program.

The group of mentors were divided into 9 groups from 9 different regions in Norway, with 2–5 mentors in each; each group had the responsibility for 5 to 10 coaches in each region. Each mentor had the responsibility for 1 to 4 coaches in the program and helped and supported the coaches individually. The mentors had four team gatherings led by a superior mentor together with their coaches, where the primary focus was the sharing of experience and competence between them. This was based on person-centred values [56]. The mentors were instructed to hold at least 10 individual consultations with their coaches, based on the principles in their own mentoring education, and participate in observations of their coaches in training and competition situations. Thus, the coach education program was based on athlete-centred values. The focus was on the social process of coaching such as how to communicate to empower the athlete and use the athletes’ own experiences in the development process. The program in the current study was therefore defined as a formal mentoring program as it was sanctioned, managed and structured by an organization [11].

### 2.3. Participants in the Pre-Test/Post-Test Control Group Design

There was an open invitation to coaches from a variety of both teams and individual sports from all parts of Norway to apply. The target group for this project was young and talented coaches that coached junior elite athletes. To be selected to participate in the program they had to be recommended by their sports federations and preferably be 30 years or younger. Every coach who applied for the program had to have a reference recommendation from their sports federation, and each coach’s sports federation had to rank coaches who applied to their own federation. A project manager in the Olympic committee ranked coaches from all sports federations based on their ranking and their application. Initially, 185 coaches applied, and out of the 109 coaches were selected, 107 accepted the invitation to participate. All the coaches were responsible for a group of athletes. In total, 734 athletes were invited to participate in the study, and 443 accepted the invitation.

A quasi-experimental intervention with a wait-list control group condition was used to investigate the effects of a one-year coach education intervention based on a formal mentoring program. The program was arranged to educate approximately 50% of the coaches the first year (from January 2019), while the other 50% would serve as a control group. The applications had one question where coaches were asked if they wanted to start in the coach education program in the first or the second year so that the coaches in the control group had agreed to be put “on hold” for 2019. In the second year, the roles were exchanged (from January 2020). This article reports the findings from the first year of the program.

A total of 66 coaches (33% women), whose ages ranged from 27 to 48 years (M = 39.58, SD = 3.55) was assigned to the experiment group. Their experience as coaches ranged from 1 to 19 years (M = 8.92, SD = 3.66). The 295 youth athletes (48% girls) that were assigned to the experimental group practised a variety of both team and individual sports (*n* > 30). Nearly half of the sample practised either cross-country skiing (16.7%), soccer (12.1%), handball (7.6%), biathlon (6.4%) or ice skating (6.1%). Among these, 15% of the athletes competed at an international elite level, 76% were considered future top-level athletes, and 9% of the athletes were engaged in recreational sports. The control group consisted of 41 coaches (32% women) whose ages ranged from 31 to 48 years (M = 41.32, SD = 3.73). Their experience as coaches ranged from 1 to 18 years (M = 7.00, SD = 4.09). The 148 youth athletes (53% girls) that were assigned to the control group practised among the same team and individual sports (*n* > 30) that the experiment group, with over half of the sample competing in cross-country skiing (14.6%), biathlon (14.6%), soccer (14.6%) and swimming (12.2%). Among these, 10% of the athletes competed at an international elite level, 71% were considered future top-level athletes, and 19% of the athletes were engaged in recreational sports. From the 107 coaches that started the program and participated in the pre-test (T1), 94 completed the data collection in the post-test (T2) (88% response rate). The coaches in the sample consisted of 65% males and 35% females. Of the 443 athletes who participated in the pre-test (T1), 228 completed the post-test (T2) (52% response rate). The athletes in the sample consisted of 51% males and 49% females. See the flowchart describing the coaches and athletes flow throughout the study in Figure 1.

The informants were told that the study was based on voluntary participation and that we appraised their consent as they handed in a completed questionnaire. The data were collected through online-based questionnaires and administrated by members of the research team. The study was conducted in accordance with the Declaration of Helsinki, and the Norwegian Centre for Research Data approved the survey (protocol code NSD-44438).

### 2.4. Instruments

In this study, we used variables that measured the coaches’ perceptions of their communication skills (attention skills and influencing skills), and both the coaches’ and athletes’ perceptions of the bond dimension of their relationship. All scales were previously validated, used in Norwegian, and measured with items answered on a seven-point Likert scale from 1 = very untrue to 7 = very true. The reliability was calculated by Cronbach’s alpha.

#### 2.4.1. Coach-Athlete Working Alliance Inventory (CAWAI)

We used a modified Norwegian translation of the Working Alliance Inventory (WAI) [57] to measure both the coaches’ and athletes’ perceptions of their relationship. WAI was originally used in a therapeutic context, In the validated sport-context adjusted version [58], words like therapist, client and therapy are altered to coach, athlete and training, respectively. The Coach-Athlete Working Alliance Inventory (CAWAI) includes three separate subscales that measure the reciprocity between the coaches’ and the athletes’ perceptions of *goals* (CAWAI-goal), *tasks* chosen to reach the defined goals (CAWAI-tasks), and their relational *bond* (CAWAI-bond). In this study, we used the CAWAI-bond dimension in the analysis. This subscale is based on four items, e.g., “There is mutual trust in the coach and athlete”, “The athlete is confident that the coach has knowledge that will be helpful”. Cronbach’s alpha for the scale was 0.76 (T1) and 0.69 (T2) for the coaches, and 0.88 in both T1 and T2 for the athletes.

#### 2.4.2. The Coach Competence Scale (CCS)

The Coach Competence Scale (CCS) [47,59] was used to measure the coaches’ perceptions of their own competence in both T1 and T2. This is a hierarchical and multidimensional scale that originally consisted of five dimensions with a total of 15 questions, including (1) Co-creating the relationship, (2) Communication, attention skills, (3) Communication, influencing skills, (4) Facilitate learning and results and (5) Make the responsibility clear. To explore whether the items measured one or more dimensions, we conducted a principal component analysis of the Coach Competence Scale (CCS) using the software package Statistical Package for the Social Sciences (SPSS) (IBM Corp. Released 2020. IBM SPSS Statistics for Macintosh, Version 27.0. Armonk, NY, USA). The Kaiser–Meyer–Olkin value was 0.78, exceeding the recommended value of 0.6 [60,61], and Bartlett’s Test of Sphericity [62] indicated statistical significance. The principal components analysis revealed the presence of three components with eigenvalues exceeding 1. However, the last component consisted of only two items with low-reliability measures, and we decided to leave it out of the further analysis. The two remaining factors explained 43.4% of the variance in total. Oblimin rotation was used to aid the interpretation and the rotated solution showed that all items had strong loadings on only one of the two components.

The first factor was labelled attention skills and consists of six items from the first two dimensions in the original scale. This explained 29.8% of the variance. Examples of statements: “In conversation with my athletes, I listen to their opinions and perspectives”, “In my coaching, I show that I understand them”. Cronbach’s alpha for the scale was 0.75/0.81 for T1/T2, respectively. The second factor was labelled *influencing skills* and consists of seven items from the third and fourth dimensions, in addition to one question from the fifth dimension from the original scale. This explained 13.6% of the variance. Examples of statements: “I ask effective and relevant questions to my athletes”, “I influence my practitioners to find solutions themselves”. Cronbach’s alpha for the scale was 0.81/0.82 for T1/T2, respectively.

### 2.5. Data Analysis

The present study adopted a classic two-wave cross-lagged panel design. At first, we used correlation analysis to describe the strength and direction of the linear relationship between the variables for both the control and experiment groups, respectively. We also included descriptive statistics of *N*, means and standard deviation. T-tests were used to compare values for each variable to elucidate possible differences between groups at T1 (baseline values) and T2. Secondly, structural equation modelling (SEM) with autoregressive cross-lagged analysis of latent (unobserved) variables was used to test our hypotheses in the Analysis of a Moment Structures (AMOS) 26 program. This analytic procedure has several advantages compared to simple multivariate tests, such as Analysis of Variance (ANOVA). Our rationale for using SEM is anchored in our three hypotheses and enabled scrutiny of several dependent variables in the same model, a combination of latent factor analysis with regression analysis, and the use of advanced statistics to adjust for missing data. We tested how observed variables (experiment participation and change in the coaches’ perception of their attention skills and influencing skills from T1 to T2) predicted changes in both the coaches’ and athletes’ perceptions of CAWAI-bond at T2. The effect of the intervention was determined using a dummy variable (0 = control group, 1 = experiment group) regressed on the outcomes controlled for by the corresponding pre-experiment measures. The coherence between observed data and the hypothesized model is reported as the goodness of fit statistics. The goodness of fit indicators used to assess the model are the Non-Normed Fit Index (NNFI, also known as TLI), Comparative Fit Index (CFI) and Root Means Square Error of Approximation (RMSEA). RMSEA ≤ 0.07, TLI ≥ 0.90 and CFI ≥ 0.90 are considered indicators of acceptable fit [63,64]. The model was tested with the whole sample and to ensure consistency in our interpretations of the results we used the standardized z-scores of the key variables in all analyses.

## 3. Results

### 3.1. Zero-Order Correlations

Zero-order correlations between the study variables as well as *N*, statistical mean, standard deviation and t-tests with Cohens D between the control- and experiment-group for each variable at both pre-test (T1) and post-test (T2) are shown in Table 1.

Table 1 reports that all correlations between the variables from the coaches’ perspective (1–6) were statistically significant, except for the correlation between attention skills T1 and influencing skills T2, as well as influencing skills T2 and CAWAI-bond T1 for coaches’ in the experiment group. All correlations are positive. This indicates e.g., that the higher the coaches perceive their attention skills or influencing skills, the higher they perceive a positive relational bond with their athletes. However, the variables from the athletes’ perspective (7–8) do not have a statistically significant correlation with the variables from the coaches’ perspective (1–6).

We also ran a t-test and assessed Cohens D to elucidate for possible differences between groups at T1 (baseline). None of the groups had statistically different values at T1. However, the Cohens D value for CAWAI-bond (coach) T2 between coaches in the control and was statistically significant (0.57). Considering effect size, Cohen [65] claims that an effect of 0.2 is small, 0.5 is medium and 0.8 is high. This finding will be commented on in the description of the findings from the SEM Analysis.

### 3.2. SEM Analysis

The relations between the variables were further analysed by means of SEM analysis for latent and observed variables. The model had a satisfactory fit to data: CFI = 0.939, TLI = 0.913, RMSEA = 0.039, chi-square = 247.137, df = 132, *p* = 0.000. This indicates plausible associations between the constructs [63,64]. Figure 1 presents the standardized regression coefficients in a cross-lagged panel structural model testing the effect of experiment participation, and perceived changes in attention skills and influencing skills (T2–T1) on CAWAI-bond (T2) from both coaches’ and athletes’ perspectives.

Figure 2 shows that the correlation between the coaches’ and athletes’ perceptions of the relational bond is not significant. The cross-lagged analysis reveals that the latent variables CAWAI-bond at T1 positively predicted CAWAI-bond for both coaches (0.68) and athletes (0.50) at T2. However, in line with the mentioned correlation, there were no significant effects across the coaches’ and athletes’ perceptions of CAWAI-bond from T1 to T2.

Furthermore, the analysis of the observed variables revealed that experiment participation significantly predicted the coaches’ experience (0.29), but not the athletes’ experience of CAWAI-bond at T2. In addition, participation in the experiment did not correlate with the coaches’ experience of change in either their attention skills or their influencing skills. The coaches’ experience of the attention skills change from T1 to T2 was significantly predicted by the coaches (0.20), but not the athletes’ experience of CAWAI-bond at T2. The coaches’ experience of the influencing skills change from T1 to T2 had no significant association with either the coaches’ or the athletes’ perception of CAWAI-bond at T2.

## 4. Discussion

The primary goal of the current study was to explore possible effects from the one-year coach education program based on formal mentoring of coaches’ and athletes’ perceptions of their relational bonds, and if they agree on their perceptions of their relational bonds. The study also investigates if changes in the coaches’ perceptions of their attention skills and influencing skills affect both coaches’ and athletes’ perceptions of their relational bonds at the post-test. The results in the current study indicate that the coach education program had positive effects on the coaches’ perceptions of their relational bonds with their athletes, but no positive effects were found among the athletes’ perceptions. The results also found that there were no significant associations between changes in the coaches’ perceptions of their attention skills and influencing skills and their athletes’ perceptions of their relational bonds with their coaches at the post-test. Thus, the hypotheses in the current study were only partly confirmed.

The first hypothesis in the current study predicted that the coach education program would positively affect both coaches’ and athletes’ perceptions of their relational bonds at the post-test (T2). The results were that no significant associations were found in the athletes’ perceptions of their relational bond with their coaches. Thus, the result indicates that the formal coach education program based on mentoring had no significant effects on the athletes. However, the standardized regression coefficients in the cross-lagged panel structural model (Figure 1) indicate significant effects on the coaches’ perceptions of their relational bonds with their athletes at the post-test (T2). This finding supports previous research [4,5] and underlines the need for further research on how different content and methods in coach education programs affect the coach-athlete relationship from both the coaches’ and their athletes’ perspectives.

The second hypothesis in the current study predicted that there would be common perceptions between coaches and their athletes about the bond dimension in their coach-athlete working alliances at both the pre-test (T1) and the post-test (T2). The results in the current study show that no such associations were found: both the zero-order correlations (Table 1) and the standardized regression coefficients in the cross-lagged panel structural model (Figure 2) found no significant associations. Thus, coaches’ perceptions of their relational bonds with their athletes did not correspond with their athletes’ perceptions of their relational bonds with their coaches. This is an interesting and somewhat surprising finding that contrasts with previous research [27,38]. However, based on the mixed findings in previous research, our findings support other studies that found significant differences between coaches’ and athletes’ perceptions [41,42]. A potential explanation might be that the athletes in the coach-athlete relationships were not involved in the development process of their coaches. The results indicate that there is no mutual understanding about the relational bonds between the coaches and their athletes and coach-education programs should consider including athletes’ perspectives about the training needs of their coaches and involve the athletes in the development process. This exemplifies the complexity of interpersonal relationships and illuminates the need for further research on the reciprocity of the coach-athlete relationship.

The third hypothesis in the current study predicted that the coaches’ perceptions of changes in their attention skills and influencing skills from the pre-test to the post-test (T2-T1) would positively affect the coaches’ and athletes’ perceptions of their relational bond at the post-test (T2). The results were that the coaches’ perceptions of changes in their attention skills positively predicted their perceptions of their relational bonds with their athletes at the post-test (T2), however, not their influencing skills. This finding supports previous research which claims that there are associations between attention skills and trustful coach-athlete relationships based on strong bonds [46,51]. However, the results from the standardized regression coefficients in the cross-lagged panel structural model (Figure 2) found no significant associations between changes in the coaches’ perceptions of their attention skills and influencing skills and their athletes’ perceptions of their relational bonds with their coaches at the post-test (T2). The results from the zero-order correlations (Table 1) also confirm this result, whereas no significant correlations were found between attention skills and influencing skills and the athletes’ perceptions of their relational bonds with their coaches. The results are rather surprising considering previous research [47,48] and questions whether the coaches trained their communication skills enough during the experiment, especially the combined use of attention skills and influencing skills. This finding calls for further research on the association between interpersonal communication skills and the coach-athlete relationship.

The results indicate that coaches are overly positive about their development as coaches relative to their athletes’ perceptions of their development. Earlier studies also support this [66,67]. A potential explanation might be that the coaches did not include their athletes’ perspectives about the coaching needs they should develop as coaches and in what area. This could improve their relational bonds with their athletes. The formal mentoring program and the learning process between the mentors and the coaches, highlighted the importance of empowering the helper in the relationship (mentor-coach/coach-athlete) through the use of attention skills such as open questioning and active listening skills. Empowerment includes focusing on the helper’s experiences and perspectives in the learning process, and the question is to what degree did the mentors in the coach education program influence the coaches’ perspectives about their relational bonds with their athletes? Awareness about developmental needs are essential to achieve changes and influencing skills are often needed to influence thought and actions of the other part in the relationship. However, the combined use of attention skills and influencing skills is challenging in the work to establish trustful relationships.

As previous studies have found, trust is the key to effective relationships, and attention skills are essential to stimulate trust. When influencing skills are used they might challenge the bond in the relationship because new perspectives are stimulated and explored, and the importance of being heard and understood in the relationship might thus be challenged. Communication is challenging and the results make us question if the coaches practised their communication skills enough, especially the combined use of attention skills and influencing skills. Communication is a practical skill and exposure to practical training is the key to develop both attention skills and influencing skills. However, this effect was not significant in the cross-lagged panel structural model (Figure 2).

Our findings lead us to question whether the mentors challenged the coaches sufficiently through the use of influencing skills in the development process. Other research has critiqued and challenged the conception of mentorship as always being beneficial for sports coaching [16,68]. This has underlined the need for explorations of the wider nuances of mentorship (e.g., how mentee coaches perceive, interpret and act on feedback from their mentors [12,69]. In a wider perspective, the literature seems to struggle to reconstruct and imagine what transformative mentoring practice for coaches entails and there is little evidence that connects mentoring to IB; a change in coaching practice [16,67]. Our results support these claims and have proposed suggestions for further research.

This study has some limitations. Future studies should use more extensive longitudinal and experimental designs with randomized controlled trials to test the development of relational bonds between coaches and athletes over time, and the effect of formal mentoring in coach education programs. In addition, all data were based on coaches’ and athletes’ self-reports. It seems appropriate and expedient to employ more objective measures of social dynamics and triangulate the subjective perceptions in the self-reports by the use of observations or other data collection methods. Furthermore, this study has only measured individual-level factors. Future studies should include factors at the group- and organizational levels through multilevel analysis to scrutinize how these systemic variables could influence the effect of coach education programmes.

## 5. Conclusions

The results in the current study indicate that there are considerable needs for further research on both the effects of formal mentoring in coach education programs, the reciprocity of the coach-athlete relationship and the development of interpersonal knowledge. The present study explores and provides insights into the understudied phenomena of reciprocity in coach-athlete dyads and the effects of formal mentoring in coach education programmes. Overall, the current research demonstrates further needs for research and underlines the complex and uncertain nature of interpersonal relationships. The key implication from the results of this study is that formal coach education programs should target reciprocal perceptions of the coach-athlete relationships as well as coach-centred skills and strategies to establish, maintain, develop and repair relational bonds with the athletes. There is a dearth of research that examines the reciprocity in coach-athlete relationships and the effects of coach educations programs. Future research should use both quantitative designs such as randomized controlled trials with control groups and qualitative designs with interviews, diaries and observational techniques to scrutinize these phenomena further and deeper.

## Figures and Tables

**Figure 1 sports-09-00116-f001:**
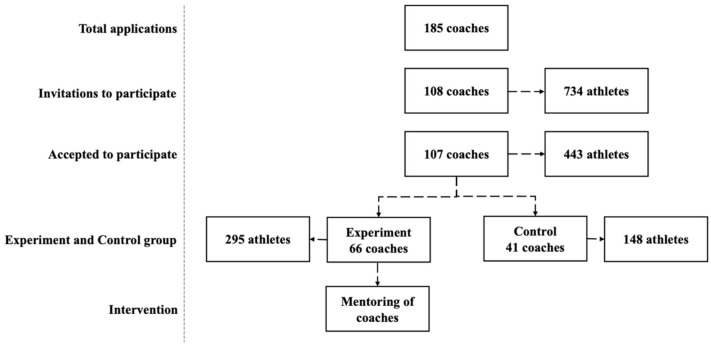
Flowchart describing the coaches’ and athletes’ flow throughout the study.

**Figure 2 sports-09-00116-f002:**
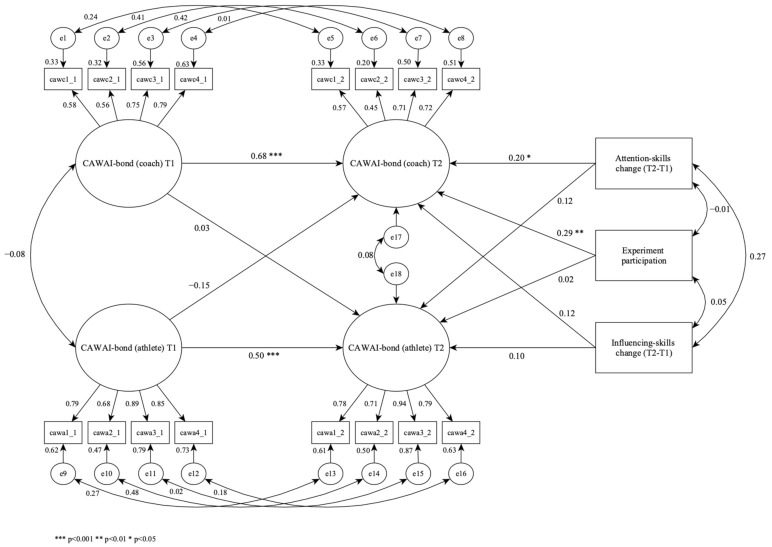
Standardized regression coefficients in a cross-lagged panel structural model testing the effect of experience participation and changes in attention skills and influencing skills (T2–T1) on CAWAI-bond at T2 from both coaches’ and athletes’ perspectives.

**Table 1 sports-09-00116-t001:** Zero-order correlations, *n*, mean, standard deviation, t-test, significance and Cohens D for control and experiment group.

Variables	1		2		3		4		5		6		7		8	
	Control	Experiment	Control	Experiment	Control	Experiment	Control	Experiment	Control	Experiment	Control	Experiment	Control	Experiment	Control	Experiment
Attention-skills (coach) T1	-															
2. Attention-skills (coach) T2	0.62 **	0.63 **	-													
3. Influencing-skills (coach) T1	0.40 **	0.44 **	0.45 **	0.51 **	-											
4. Influencing-skills (coach) T2	0.50 **	0.21	0.65 **	0.50 **	0.66 **	0.72 **	-									
5. CAWAI-bond (coach) T1	0.49 **	0.52 **	0.39 *	0.41 **	0.47 **	0.29 *	0.57 **	0.10	-							
6. CAWAI-bond (coach) T2	0.47 **	0.31 *	0.49 **	0.49 **	0.51 **	0.29 *	0.59 **	0.37 **	0.73 **	0.57 **	-					
7. CAWAI-bond (athlete) T1	−0.01	0.10	−0.14	0.19	−0.20	0.22	−0.35	0.24	−0.29	0.11	−0.50 *	0.12	-			
8. CAWAI-bond (athlete) T2	−0.05	−0.05	−0.01	0.19	0.26	−0.10	0.14	0.17	−0.07	0.08	−0.25	0.08	0.57 **	0.44 **	-	
*N*	42	65	37	59	42	65	37	59	42	65	37	58	148	295	75	148
Mean	6.18	6.18	6.20	6.18	5.50	5.56	5.60	5.70	5.64	5.70	5.57	5.89	6.06	6.14	5.93	6.05
SD	0.50	0.50	0.53	0.51	0.57	0.57	0.59	0.54	0.69	0.62	0.56	0.56	0.96	0.99	1.20	0.93
t	−0.04		0.12		−0.55		−0.90		−0.48		−2.70		−0.86		−0.79	
p	0.97		0.90		0.59		0.37		0.63		0.01 **		0.39		0.43	
d	-		-		-		-		-		0.57		-		-	

Note: Abbreviations: CAWAI = Coach-Athlete Working Alliance Inventory ** *p* < 0.01 * *p* < 0.05.

## Data Availability

The data presented in this study are available on request from the corresponding author.

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
