# Peer review of "Effects of a Mentor Program for Coaches on the Coach-Athlete Relationship"

_sports, 2021, doi:10.3390/sports9080116_

Round 1
Reviewer 1 Report
Please find my specific comments and recommendations in the attached file.

Author Response
Dear Sirs,
Related to our submitted manuscript entitled “Effects of a Mentor Program for Coaches on the Coach-Athlete Relationship: A One-Year Control-Experiment Study Including Data from Coaches and Athletes”, submitted to Sports, we received the several constructive comments from the reviewers. We carefully read and interpreted the feedback that enabled us to assess the article with a fresh glance.
In the following pages we will present our revisions in tables. The left column describes the reviewers` comments, and the right column describes our revisions based on these comments.
We hope that the comments from reviewers are interpreted in a satisfactory manner, and that the revisions based on these are considered adequate. The editing and revising work with the manuscript based on the comments from the reviewers, contributed in our opinion to a text at a higher level of quality and we hope the reviewers agree.
Best Regards,
The authors

Reviewer 2 Report
I would like to commend the authors on an interesting manuscript that will add to the coach-athlete relationship literature. Overall I find the manuscript well written and insightful. The following comments are provided to help enhance the manuscript further.
Abstract:
The opening sentence is awkwardly written. Please revise to give a good introduction into your study.
In the fourth sentence 'by a mentor' could be removed. It can be assumed that mentoring is provided by a mentor.
Introduction:
The section before the heading 'The coach-athlete relationship' needs to be revised. Currently it does not introduce the topic area and adds little. I would suggest an opening paragraph that provides a justification for this study is needed. Why is this important and what is missing from the current literature?
Method:
Is it possible to provide more detail on what other education did the participating coaches get? Did they just receive mentoring or was this part of a larger education program?
Discussion:
While reading this I wondered if the coach's perceptions of the relationship had changed from their original evaluation i.e., do they think they may have over-scored the relationship before the mentoring program? By providing mentoring their expectations and understanding will have changed. It may not be possible to address this in the current study.
L351 'coaches perceptions of their relational bonds...' - This is a very interesting observation and is worth exploring in more detail. Could the authors add further to this?
L411 - Again this is an important discussion regarding the use and effect of mentors. It would be nice for the authors to articulate this further
Author Response

(The authors gave the same response as above.)

Reviewer 3 Report
Manuscript ID: sports-1278428
The manuscript is good, well structured, wide sample, ambitious program and interesting topic. The final results are not as good as they expected, but the resarch is worth it to be done. Maybe the control of the program (too many coaches) is one of the reasons. Some aspects to consider:
Title
- Too long. Is it really necessary the second part of it: “A One-Year Control-Experiment Study Including Data from Coaches and Athletes”. In my opinion, it is better a short one.
Key Words
- This key Word is already in the title: “coach-athlete relationship” (not necessary to repeat). Instead of it, you could add “Control-Experiment Study” if you definivily remove it from the tittle or another key word.
Introduction
- Line 23-25 This statement needs an academic reference
- Line 62 “…tionship over time), complementarity” add and “…tionship over time), and complementarity”
- Line 128 complete the objective, adding the aim of the program: interpersonal relationship and the perception of coaches and athletes.
Method
- Methodology is good enough.
- line 142 Why do you not mention NOSC in the introduction? Do you consider to introduce a bit about it in the first part of the manuscript?
Author Response

(The authors gave the same response as above.)
